# Regional uniqueness of tree species composition and response to forest loss and climate change

Nina van Tiel [1,2] ✉, Fabian Fopp [3,4], Philipp Brun [4], Johan van den Hoogen[1], Dirk Nikolaus Karger [5], Cecilia M. Casadei [6,7], Lisha Lyu[3,4], Devis Tuia [2], Niklaus E. Zimmermann [4], Thomas W. Crowther [1,8] & Loïc Pellissier [3,4,8]

The conservation and restoration of forest ecosystems require detailed knowledge of the native plant compositions. Here, we map global forest tree composition and assess the impacts of historical forest cover loss and climate change on trees. The global occupancy of 10,590 tree species reveals complex taxonomic and phylogenetic gradients determining a local signature of tree lineage assembly. Species occupancy analyses indicate that historical forest loss has significantly restricted the potential suitable range of tree species in all forest biomes. Nevertheless, tropical moist and boreal forest biomes display the lowest level of range restriction and harbor extremely large ranged tree species, albeit with a stark contrast in richness and composition. Climate change simulations indicate that forest biomes are projected to differ in their response to climate change, with the highest predicted species loss in tropical dry and Mediterranean ecoregions. Our findings highlight the need for preserving the remaining large forest biomes while regenerating degraded forests in a way that provides resilience against climate change.

The UN Decade on Ecosystem restoration has begun to catalyze interest in nature restoration, from the conservation and management of existing forests to the rejuvenation of ecosystems on degraded lands[1,2]. Sustainable management offers an opportunity to address biodiversity loss and climate change, and at the same time enhance human well-being across the globe[3]. A key component in this global effort is the conservation and restoration of forests, which represent the largest repositories of biodiversity and carbon on land[4,5] and are associated with essential ecosystem services[6]. Earth is home to ~73,000 tree species[7] and understanding the spatial distribution of these species is a challenge to be tackled to guide regional to global

management policies. Conservation and restoration efforts should be guided by solid information on species' ecological preferences, taxonomic and phylogenetic diversity, and ecosystem resistance to climate change[8].

The latest assessment indicates that at least 31% of all tree species are threatened with extinction globally[9]. This loss of biodiversity has far-reaching consequences[10], and halting and reverting it requires the identification of priority conservation areas to target regions of greatest threat. The selection of such regions may consider different facets of biodiversity, such as taxonomic, functional, and phylogenetic diversity, in an effort to prevent the extinction of species, as well as

[1]Global Ecosystem Ecology, Department of Environmental Systems Science, ETH Zürich, Zürich, Switzerland. [2]Environmental Computational Science and Earth Observation Laboratory, Ecole Polytechnique Fédérale de Lausanne, Lausanne, Switzerland. [3]Ecosystems and Landscape Evolution, Institute of Terrestrial Ecosystems, Department of Environmental Systems Science, ETH Zürich, Zürich, Switzerland. [4]Land Change Science Research Unit, Swiss Federal Institute for Forest, Snow and Landscape Research, WSL Birmensdorf, Switzerland. [5]Biodiversity and Conservation Biology, Swiss Federal Institute for Forest, Snow and Landscape Research, WSL Birmensdorf, Switzerland. [6]Laboratory of Biomolecular Research, Biology and Chemistry Division, Paul Scherrer Institute, PSI Villigen, Switzerland. [7]Institute of Molecular Biology and Biophysics, Department of Biology, ETH Zürich, Zürich, Switzerland. [8]These authors jointly supervised this work: Thomas W. Crowther, Loïc Pellissier. ✉e-mail: nina.vantiel@epfl.ch

unique branches of evolutionary history[11,12]. The strongest gradients in tree species composition are often observed at the boundary between major plant kingdoms following deep historical splits[13], or between juxtaposed biomes, such as between tropical and temperate forests[14], associated with rapid changes in climate, topography, and other environmental conditions[15]. These major shifts in tree lineage composition are complemented by gradients of community composition within biomes, influenced by a complex interplay of abiotic factors at regional scales, such as variations in freezing frequencies or moisture[16]. A better understanding of these gradients is not only important to comprehend the conditions sustaining biodiversity hotspots but also for conserving and restoring forest ecosystems and ensuring their continued contribution to biodiversity and ecosystem services. A few studies have investigated tree species composition at the global scale, using only the location of observations to construct species' ranges[17] or to compute species richness[18]. Basing such analyses on habitat suitability maps may allow a better estimate of diversity in regions with limited observational data. However, suitability-based investigations of tree species composition have so far been limited to a relatively small number of tree species or local to regional scales[19,20]. A global high-resolution mapping effort would offer a picture of the main gradients of taxonomic and phylogenetic composition in the forests from local to global scales.

Establishing effective conservation targets requires a sound understanding of the historical decline of species ranges associated with habitat loss[21]. For instance, Hubbell et al. used neutral theory to estimate how the decline of forest cover could have impacted tree species diversity[22]. However, the distribution of tree species is regulated by non-neutral ecological determinants[23], and the consequence of habitat loss could be informed by estimated species occupancy[24]. In particular, species potential ranges can be matched with terrestrial land cover to map the species' area of habitat (AOH, also known as the extent of suitable habitat), which is considered a criterion for establishing species ranking on the IUCN red list[25]. However, in contrast to vertebrates[26], the estimated species ranges for trees are limited to selected families[27] or available at coarse resolution only[17], which limits assessments of the effect of habitat loss on tree diversity at the regional scale. To overcome this challenge, machine learning can support the mapping of species' habitat suitability with high predictive power[28]. Then, combining these maps with data on forest cover estimated from remote sensing[29,30] can provide insights into historical range loss and, therefore, support management decisions for conservation[31].

The challenges of forest management are significantly compounded by the rapid changes in climate[32]. As climatic conditions change, the potential distribution of plant species will also shift[33,34], leading to changes in species composition[14] and a potential decline in provided ecosystem services[35]. Moreover, the response of species to those changes can largely vary across regions, where some might suffer more extinctions, while others might be able to track climate change along latitude or elevation[36]. Thus, climate change-resilient forest and ecosystem management require an understanding of which species have the potential to thrive under future climatic conditions. However, predicting the potential distribution of tree species under changing climate conditions is a major challenge as model predictions incorporate various limitations and assumptions[37,38]. Nevertheless, estimating future suitable ranges with statistical models remains a scalable approach to gain insights into the effects of climate change and inform conservation and restoration efforts.

The objective of this study is twofold: first, we estimated the spatial distribution of a large number of tree species at the global level at a 30-arc second resolution. We developed a cloud-based implementation of an environmental niche modeling pipeline with geographic dispersal constraints and over 26 million recorded observations of trees from a broad compilation of databases (Table 1) to generate range maps for 10,590 tree species for which we had at least 90 occurrence records, which was found to be sufficient to train models with good predictive performance. Subsequently, and beyond the pure large-scale mapping effort, we used the predicted distributions to perform three analyses investigating the global distribution of tree species. First, we inferred global gradients of species composition and investigated the global variation of taxonomic and phylogenetic forest composition. Second, by combining potential species occupancy with tree cover data, we computed the reduction in species ranges when constrained to current forest cover, therefore assessing the impact of historical forest loss on species range reduction. Third, using climate change simulations, we investigated the extent of species turnover and shifts in latitude and elevation under projected climate change. This study provides a comprehensive overview of the global forest composition, the characterization of species' range sizes, and the resilience of tree biodiversity to climate change. Thanks to large publicly available occurrence databases and our cloud-based implementation, we are able to reach results of unprecedented scale for habitat suitability mapping, both in terms of spatial coverage and resolution and in terms of the number of species, allowing us to investigate the global composition of tree species at very high granularity in our subsequent analyses.

## Results
### Model performance
We applied a model combining geographic range polygons and species distribution modeling to map the ecological suitability of tree species, based on climatic and pedologic variables, at a 30-arc second resolution. Observational data was collated from 13 databases (Table 1) and aggregated to the same resolution as the model covariates. Geographic range polygons were constructed, using reported native countries and the location of occurrences, as coarse estimated of species' native ranges. They were used to filter observation records and to constrain the extents of the projected ranges. We modeled the distributions of 24,140 tree species with at least 20 distinct observations after aggregation, and with reported native countries available from GlobalTreeSearch[39]. We assessed the performance of the models and found our modeling approach to obtain sufficient predictive performance for species with at least 90 spatially explicit observations (Supplementary Fig. 1). In total, 10,590 tree species met this criterion. When evaluated by 3-fold cross-validation, the models achieved an average true skill statistic (TSS) of $0.77 \pm 0.11$ and an average area under the ROC curve (AUC) of $0.93 \pm 0.04$ (Supplementary Fig. 2). The evaluation of the models on an independent presence-absence dataset from sPlot[40] yielded lower TSS ($0.53 \pm 0.30$) with a high recall and low precision (Supplementary Fig. 3). This points towards a low binarization threshold, which is expected for low-prevalence species when optimizing the threshold on maximum-TSS. With an average false positive rate of 29.5%, false positive errors were the most common than false negative errors which had an average rate of 17.3%. False positive errors, where the model classified a location to be suitable for a species although no occurrence was recorded there, are not necessarily problematic as the species may be absent due to factors that were not considered by the model, such as biotic interactions, human influence, or stochasticity. Moreover, the comparison of our modeled distributions to maximum habitat suitability maps for 23 European species for which confident estimations were available in a published report[41] showed some considerable differences with an average intersection over union (IoU) of 0.42 (Supplementary Fig. 4). However, our models were generally more conservative with ranges on average 36% smaller than those to which they were compared. Unlike previous large-scale modeling efforts[42], our approach combined both species distribution modeling and geographic range limits, which can, for instance, represent dispersal limitations[43] and consider reported native ranges.

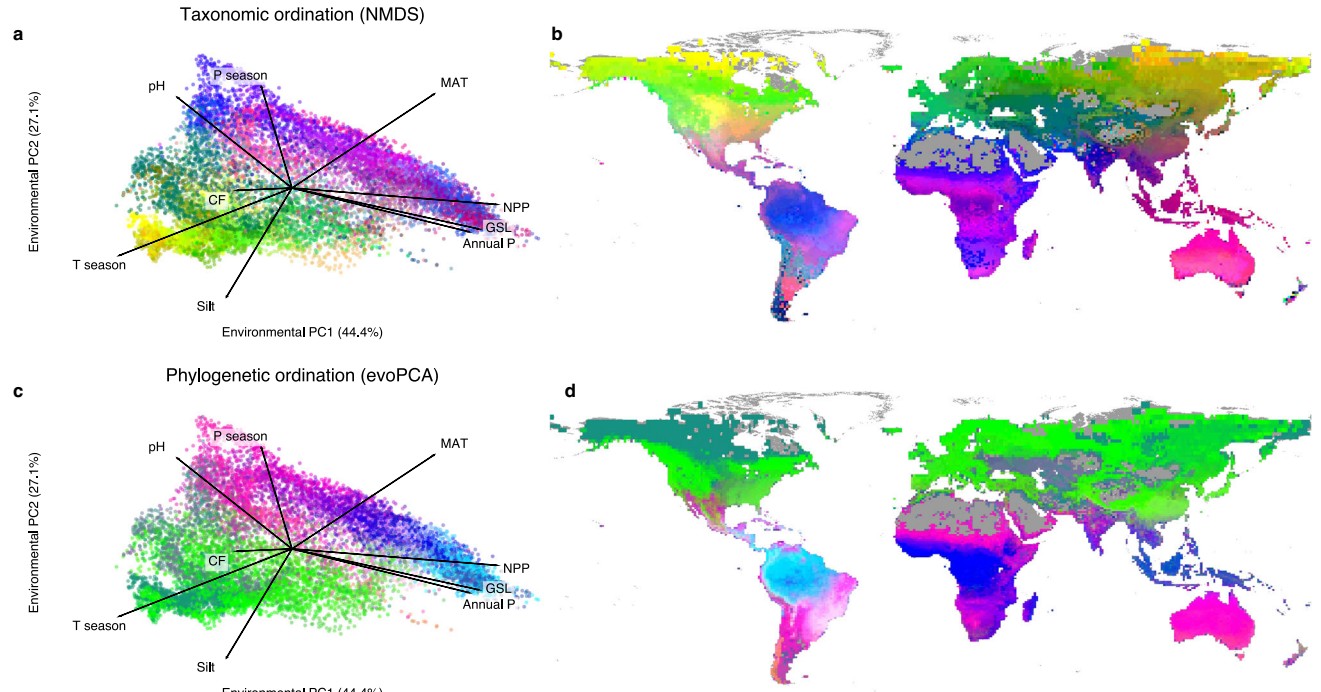

**Fig. 1 | Gradients in taxonomic and phylogenetic composition show a near-unique biodiversity signature of every single location on the planet.** Taxonomic composition is represented by a 3-axis non-metric dimensional scaling (NMDS) and phylogenetic beta-diversity is represented by the 3 first axes of a phylogenetic ordination (evoPCA). Both the taxonomic and phylogenetic ordinations are computed on the global community matrix derived from the modeled distributions of $n = 10,590$ tree species sampled at a resolution of 100 km, resulting in $n = 12,548$ sites. The 3 axes of each ordination are mapped to red, green, and blue with minimum and maximum values corresponding to the 10th and 90th percentiles.

**a**, **c**. Scatter plot of taxonomic and phylogenetic ordinations in environmental space, a 2-dimensional space made up of the 2 first axes of a PCA of the environmental variables used for species distribution modeling: mean annual temperature (MAT), temperature seasonality (T season), annual precipitation (Annual P), precipitation seasonality (P season), growing season length (GSL), net primary productivity (NPP), silt content (Silt), coarse fragments (CF), and soil pH (pH). **b**, **d**. Map of taxonomic and phylogenetic ordinations in geographical space. Source data are provided as a Source Data file. The maps were created with QGIS[110] and the gray base map corresponds to all areas for which model predictors were available.

## Unique species composition across the globe

Our models show that global tree taxonomic and phylogenetic composition is organized in complex biogeographic and environmental gradients worldwide. By computing non-metric multidimensional scaling (NMDS) on the community matrix based on our predicted distributions, sampled at using an equal area projection at a resolution of 100 km, we can visualize an estimate of tree species composition globally (Fig. 1a, b and Supplementary Fig. 5a–c). Combining the community matrix with a phylogeny of the 10,590 species considered[44], we computed a phylogenetic ordination[45,46] thus obtaining an estimate of phylogenetic beta-diversity globally (Fig. 1c, d and Supplementary Fig. 5d–f). Taxonomic and phylogenetic composition show marked spatial gradients, where any given locations displays a unique signature in taxonomic and phylogenetic composition. Species composition also varies in environmental space, with similar taxonomic or phylogenetic compositions often found under similar environmental conditions, although we also observe dissimilar compositions under similar environmental conditions, such as across tropical moist forests.

Taxonomic turnover is described by smooth distributions both in geographical and ordination space (Supplementary Fig. 5c). The lack of structure in the underlying distributions can be characterized by the difficulty of finding an adequate clustering: we found that two clusters obtained the best score, albeit with high intra-cluster variance and massive cluster size (Supplementary Fig. 6a, b). Phylogenetic turnover showed clearer spatial boundaries and more peaky distributions (Supplementary Fig. 5f). One of the most striking peaks of these distributions corresponds to a very homogeneous phylogenetic composition across Northern Canada and Alaska which also corresponds to

one of the six clusters formed by the phylogenetic ordination (Supplementary Fig. 6c, d). Redundancy analysis indicates that gradients in phylogenetic beta-diversity are associated with shifts in climatic and edaphic factors, with 57.6% variance explained by the nine variables used for niche modeling. In contrast, the variation in taxonomic diversity was not well captured by the same abiotic variables, with only 11.8% variance explained. For both ordinations, variation partitioning analysis indicated that climatic variables had stronger effect than edaphic variables (Supplementary Fig. 7). This may be associated to scale-dependency or to the soil data being generated by models reliant on climate-related variables[47]. Together, the lack of structured distributions and of fine-grained clusters in these ordinations point towards a near-unique biodiversity signature, as any given site is associated with a unique set of tree species and lineages resulting from a combination of historical and ecological factors.

## Potential species occupancy restricted by tree cover

Combining remotely sensed tree cover and the potential habitat suitability of species described above, we documented that the suitable occupation area, considering ecological preference together with biogeographic and dispersal limitations, is significantly restricted by forest cover loss. We computed species' range sizes based on their modeled distributions, and range sizes of the distributions constrained to forested areas, defined as areas with at least 10% tree cover as estimated using remote sensing data from the year 2000[29]. The relative decrease in range size reflects how species ranges are restricted by historical forest cover loss and we find a median range reduction of 22.0% across all considered species. All biomes showed significantly lower realized ranges compared to potential ranges of species

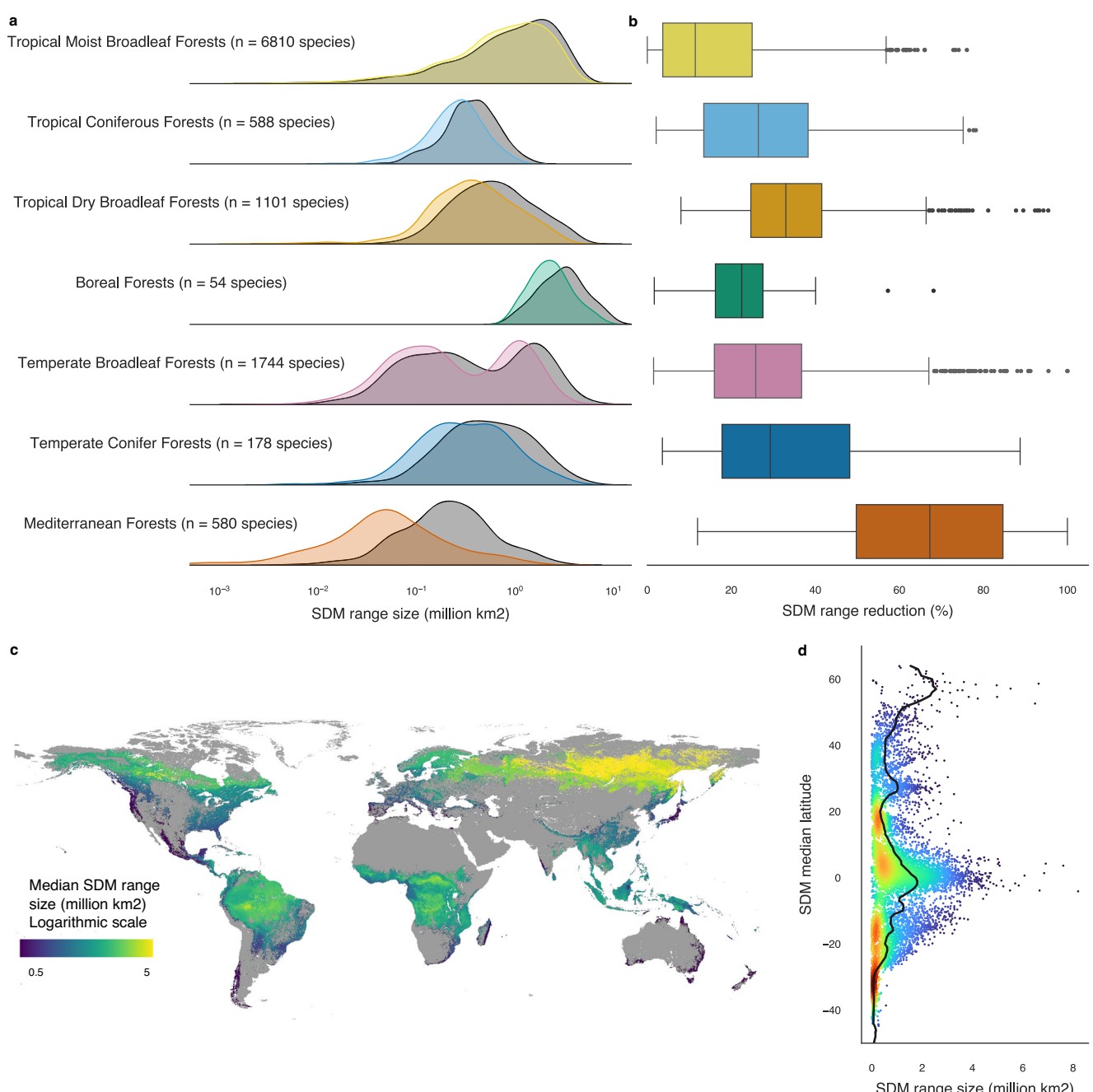

**Fig. 2 | Species occupancy range distribution and loss. a** Distributions of species occupancy range sizes globally (gray) and constrained to forests (at least 10% tree cover, color) for species in each forest biome. **b** Boxplot of relative range reduction across species in each forest biome with the center line showing the median, the box limits the quartiles, the whiskers 1.5 times the interquartile range, and the points the outliers. The distributions and boxplots are computed for $n = 6810$ species for Tropical Moist Broadleaf Forests, $n = 588$ species for Tropical Coniferous Forests, $n = 1101$ species for Tropical Dry Broadleaf Forests, $n = 54$ species for Boreal Forests, $n = 1744$ species for Temperate Broadleaf Forests, $n = 178$ species for Temperate Conifer Forests and $n = 580$ species for Mediterranean Forests. **c** Global map of median species range size constrained to forests, created with QGIS[110]. The gray base map corresponds to all areas for which model predictors were available. **d** Plot of species' median latitude against range size constrained to forests, colored by point density, where red indicates the highest density. Source data are provided as a Source Data file.

($p < 0.01$ for t-test between unrestricted and restricted range sizes over all species and per biome) with median range reduction per biome varying between 11.4% and 67.2%.

This decline was especially pronounced in the tropical dry broadleaf forests and temperate conifer forest biomes (33.0% and 29.3% median range reduction, respectively), and even more in the Mediterranean biomes with a 67.2% median range reduction (Fig. 2b). Despite increasing forest loss in the last two decades[29], some biomes had so far limited decline and those biomes still host species with

uniquely broad ranges, typically at high latitudes in the boreal forests, but also around the equator in the Amazon, Congo basin and South East Asian forests (Fig. 2c and Supplementary Fig. 8). While all species found in boreal forests were predicted to have large ranges, the ranges of species in tropical moist forests have a large spread in range size, spanning three orders of magnitude. For instance, the species with the smallest (*Palaquium sericeum*) and the largest (*Terminalia grandis*) range sizes with no tree cover restriction are found in the tropical moist broadleaf biome.

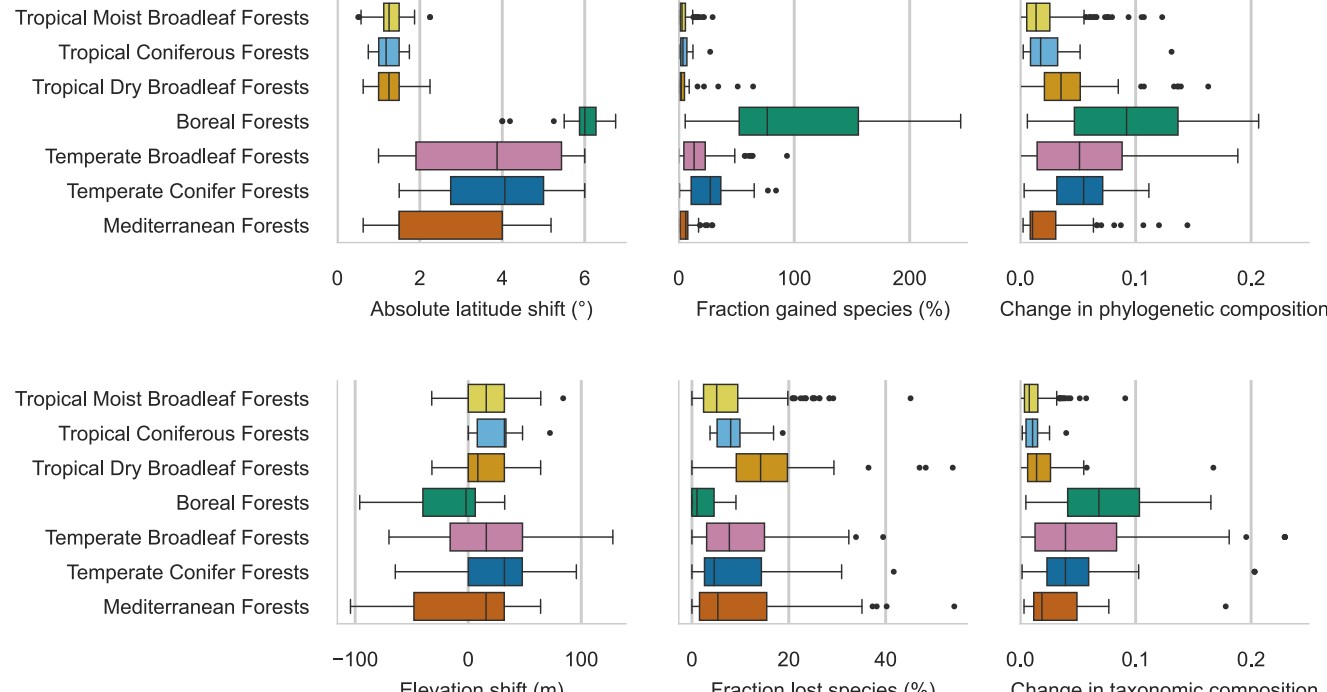

**Fig. 3 | Response of tree species to climate change across biomes.** The median absolute latitude and median elevation shift among species, fraction of lost and gained species, and change in taxonomic and phylogenetic composition under climate change were computed for each forest ecoregion. The boxplots show statistics for $n = 239$ ecoregions for Tropical Moist Broadleaf Forests, $n = 14$ ecoregions for Tropical Coniferous Forests, $n = 55$ ecoregions for Tropical Dry Broadleaf Forests, $n = 26$ ecoregions for Boreal Forests, $n = 91$ ecoregions for Temperate Broadleaf Forests, $n = 49$ ecoregions for Temperate Conifer Forests and $n = 61$ ecoregions for Mediterranean Forests. The center line of the boxplots shows the median, the box limits the quartiles, the whiskers 1.5 times the interquartile range, and the points the outliers. Changes are computed between predicted distributions with climate variables for 1981-2010 and climate projections for 2071-2100 under climate change scenario SSP 5.85. Changes in composition are computed as the Euclidean distance between scaled NMDS and evoPCA values computed at the ecoregion level. Source data are provided as a Source Data file.

## Response of tree species to climate change across biomes

We compared species distributions mapped with climatic variables averaged over 1981-2010 with distributions with projected future climate for 2071–2100 with emissions corresponding to the shared socioeconomic pathway (SSP) 5.85. Predicting future species distributions has many limitations[37], yet it presents a scalable approach to grasp how climate change may affect species distributions in the future. Acknowledging that there are many uncertainties at the species level, we investigated trends at the scale of regional ecosystems, using the RESOLVE ecoregions[48]. The results for each of the 497 forest ecoregions were then aggregated to their corresponding biome (Fig. 3). Our results predict that ecoregions in different biomes will show significant diverging behavior in their response to climate change (Pillai's Trace test, $p < 0.001$).

Species distributions in all forest ecoregions were predicted to undergo latitudinal shifts, with the smallest shifts found in tropical forest ecoregions, with a median of 1.25°, and particularly pronounced shifts in boreal forests with a median of 6.00°. Elevation shift presented a larger variation across ecoregions, with a coefficient of variation of 3.07, compared to 0.73 for shifts in absolute latitude, pointing towards a higher variability of available habitat in elevation than in latitude. We find relatively small median elevation shifts in most forest ecoregions, with an absolute median shift of less than five meters in about 24% of the considered ecoregions. Temperate and tropical coniferous forest biomes presented the highest positive median elevation shifts across ecoregions, with medians of 32 meters across ecoregions in both biomes. However, some ecoregions were estimated to have large downward median elevation shifts, in particular in the boreal forest biome with a median elevation shift of −1.95 meters. While most species are expected to shift their ranges upwards, downward elevation shifts have been reported in multiple studies[49,50] and may reflect both newly available suitable habitats and changes in precipitation regimes[51].

Our results indicate that, in boreal forest ecoregions, many new species may find suitable habitats and few species may lose suitable habitats under future climate change. This is coherent with our predictions of larger shifts in latitude or elevation allowing species to track climate change thanks to continuously available habitat and is estimated to have a strong effect on the phylogenetic composition of these ecoregions. On the other hand, tropical dry and Mediterranean forests, with generally much more geographically restricted species' ranges, were found to suffer the highest proportion of extirpated species, with some ecoregions predicted to lose suitable habitat for over 50% of currently suitable species. In Mediterranean ecoregions, the turnover of species is estimated to affect taxonomic composition more strongly than the phylogenetic composition. Overall, more modest changes in composition were found in tropical forest ecoregions than in temperate and boreal forest ecoregions.

## Discussion

Understanding tree species distribution is central to the conservation, restoration, and management of global forest biodiversity. Here, we provide species occupancy estimations for over 10,000 tree species, beyond regions where species range maps are already available and broadly used by foresters to manage forest resources[52]. Our analysis reveals the high uniqueness of local lineage composition, where any region across the globe is characterized by a unique tree composition signature associated partially with ecological factors. Residual variance may be attributed to regional and historical factors[53], such as plate tectonics and long-distance dispersal[54]. By restricting species' suitable

habitat to forests with tree cover estimated from remote sensing[29], realized tree species distribution is significantly narrower than its potential, even when considering geographic range limits. Considering predicted species distributions under future climate projections shows that different areas will be affected differently by climate change and stresses the importance of climate-smart conservation efforts that are adapted to the specific region.

Modeling species distributions for such a large number of species was facilitated by the use of Google Earth Engine, a cloud-based platform for geospatial analysis, and its available implementation of machine learning algorithms[55]. Additionally, models with adequate predictive performance can only be constructed if enough occurrences are available for a given species. This is made increasingly possible by the availability of large databases and citizen science efforts[56]. Such global databases typically display important spatial bias[57,58], although efforts to concatenate different databases, as done here, may mitigate the acquisition bias from a single dataset. Nevertheless, to ensure satisfactory predictive performance, we excluded many rare and under-sampled species and our results are likely to present biases towards common and large-ranged species. Future work may consider the recent application of deep learning to species distributions modeling, which allows the joint estimation of the distributions of thousands of species with a single model and has been shown to obtain more reliable results for species with very few occurrence records[59–61]. Moreover, the availability of global maps of environmental variables allowed us to make continent-wide predictions of species distributions. While we selected variables that are of general ecological relevance across all biomes, future work may consider more specific factors, such as the fire frequency and severity. The cloud-based data integration and modeling approach demonstrate the potential of high-performance computing infrastructure to support the documentation of global biodiversity patterns at scales and resolutions unattained before, therefore allowing fine-grained, global analyses such as those presented here.

With the distributions of thousands of tree species, we showed a striking uniqueness of local tree species and lineage composition with both large- and small-scale turnover. The phylogenetic beta-diversity showed a spatial organization matching known phytogeographic boundaries, such as the split between Gondwanan and Laurasian biotas[13], or the clustering of tropical forests[54,62,63]. In contrast, some regions displayed phylogenetic similarities despite the large geographic distance between them, which reflects historical connectivity. For instance, the phylogenetic similarity of the Indo-Pacific region may highlight the signal of its past connection before their segregation caused by plate tectonics[62], or long-distance dispersal[54]. The similarity across Palearctic and Nearctic temperate and boreal forests is likely to indicate the long history of connectivity between these regions[64]. Within these larger biogeographic boundaries, our results further highlighted the role of environmental gradients in shaping tree lineage turnover, such as the compositional divide in tree assemblages in the Americas between tropical and extratropical environments, which is related to temperature[16] and moisture gradients[65]. Taxonomic composition showed more smooth gradients and was only weakly associated with environmental variables, suggesting a strong effect of historical factors, such as geographic range limits. Focusing on the most common tree species, our analyses uncovered marked local gradients in species and lineage composition. However, the global composition of tree species and the corresponding phylogenetic diversity may be even more strongly geographically structured when including rare species.

Beyond the current habitat suitability of local species, it is important for conservation and restoration efforts to take into account the past and the future with estimates about both the historical decline of species' ranges and to understand the spatial variation in climatic risks to forests to guide climate-smart ecosystem management[66,67]. We

derived estimates of these effects from our modeled species distributions. However, it is important to acknowledge the limitations associated with our approach. To compute historical species' range reduction, we compare species' potential range size with their ranges constrained to current forests. Yet, computing potential ranges based only on ecological suitability and biogeographic limitations may lead to an overestimation of range size, as competition with other vegetation types is likely to restrict realized species distributions. Moreover, since trees may also be present outside of forests[68,69], constraining ranges to areas with at least 10% tree cover is likely to underestimate the realized range of some species. Nevertheless, the extent of the discrepancy between potential and actual distributions remains indicative of the extent of historical forest loss due to land use change[70]. To predict future species distributions under climate change, we use projections of several climate models used by the IPCC (CMIP6) which are associated with substantial spatial and temporal uncertainty[71] and do not directly account for extreme events such as floods, fires, droughts or floods which can cause large-scale forest die-backs[72]. Furthermore, our models present added uncertainties associated with limitations due to their correlative nature and the assumption that the relationships of species with environmental covariates remain constant through time. Although mechanistic models may present a better approach to predict range shifts under climate change[73,74], they are challenging to implement and require data that is not as widely available, as well as the identification of key limiting processes[38]. Therefore, correlative models remain the most feasible option to predict the effects of climate change for a large number of species and at a global scale. Finally, the bias of our results towards well-sampled species should be carefully considered. The exclusion of species for which not enough occurrence records were available is likely to have led to an underestimation of the number of species with small ranges, in particular in tropical biomes due to sampling biases. Rare and small-ranged species may also be particularly vulnerable to change and our results regarding the response of ecosystems to climate change may be affected by their omission.

Despite these limitations, our results may serve as a guide for forest stakeholders, offering a global perspective that can be combined with more local observations and expert knowledge to conserve the preserved area and to decide which tree species to consider for restoration[75]. For instance, in Mediterranean ecoregions, known for a long history of human land use and pastoralism[76], we find by far the strongest range reduction, leading to small realized species ranges and high predicted extirpation rates under climate change, highlighting the need for climate-smart restoration in these forests. In contrast, in accordance with a remote sensing-based estimation[77], we found that the tropical moist broadleaf forests had the greatest intactness. Specifically, Amazonia, the Congo Basin, and South East Asia are home to not only the highest species richness (Supplementary Fig. 9) but also some of the species with the largest ranges on Earth, alongside many small-ranged species. Our results relate to the finding of the hyperdominance of some species in the Amazon basin[78], suggesting that both extreme ecological generalism and specialism are viable strategies in tropical environments. However, small ranges may also reflect non-equilibrium dynamics rather than narrow ecological niches[79]. Outside the tropics, the boreal forest shows the broadest ranges of species with the lowest decline associated with habitat loss. Moreover, the climate of boreal forests is predicted to become suitable for many species that are currently not found there.

Our results underline the uniqueness of species composition across the globe, and also the diversity of species' historical range reduction and response to climate change, reflecting that climate change displays distinct intensities and effects globally[80]. While local forest protection efforts should be made everywhere, international efforts for forest conservation strategies should primarily focus on

**Table 1 | Online databases used as data sources for occurrence data**

| Database name | URL |
|---|---|
| Botanical Information and Ecology Network (BIEN)[104] | https://bien.nceas.ucsb.edu/ |
| BIOMASS[105] | https://www.nature.com/articles/sdata201770#data-records |
| Caudullo et al. 2017[52] | https://doi.org/10.1016/j.dib.2017.05.007 |
| CONIFER | https://herbaria.plants.ox.ac.uk/bol/conifers |
| DRYFLOR[106] | http://www.dryflor.info/data/datadownload108 |
| GBIF[107] | https://www.gbif.org/occurrence/download/0032444-200221144449610 |
| GFBI[7,18] | https://www.gfbinitiative.org/ |
| IDIGBIO | https://www.idigbio.org/portal/search |
| INDIABIODIVERSITY | https://indiabiodiversity.org/observation/list |
| MUSEUM | Field collection records and manually georeferenced herbarium data from Naturalis (L) and Paris (P) natural history museums. Unpublished data collected by J.S. Strijk. |
| PNG | http://www.pngplants.org/search.htm |
| PREDICTS[108] | https://data.nhm.ac.uk/dataset/the-2016-release-of-the-predicts-database |
| RAINBIO[109] | https://gdauby.github.io/rainbio/download_page.html |

conserving the remaining most pristine extensive forest ecoregions. The strongest effort of restoration may be performed in temperate forests where historical losses were highest and recovery might bring the most direct benefits. Furthermore, tropical dry and Mediterranean regions may require the most acute conservation attention to avoid the extinction of a large faction of species. Together, our findings emphasize the need for local restoration and conservation decisions that are tailored to the unique characteristics of each specific forest.

## Methods

We used the python API for Google Earth Engine[55] to create a pipeline for high-resolution species distribution modeling. The distribution of each species is modeled separately. After data preparation, the pipeline consisted of two parts: (1) constructing the geographic range in which the species is considered, and (2) modeling ecological suitability with an ensemble of tree-based machine learning models, allowing us to map the species distribution using current climate covariates, as well as future climate projections. Further analyses were applied to the results of our models using the python API for Google Earth Engine v0.1.329[55] with python v3.8.13[81] with helper packages: pandas v1.4.4[82], numpy v1.23.4[83], matplotlib v3.5.3[84] and seaborn v0.12.1[85]. Moreover, some analyses were performed using R v4.2.2[86] with helper packages: tidyr 1.3.0[87], tidyverse 1.3.2[88], tibble 3.1.8[89], data.table 1.14.6[90], dplyr 1.1.0[91], gridExtra 2.3[92] and ggplot2 3.4.1[93].

### Data sources and cleaning

Tree species occurrence data were downloaded from 13 different online databases, published and unpublished datasets (Table 1). We kept only records with standardized tree species names, according to the global tree checklist GlobalTreeSearch[39] (v1.3, downloaded in January 2020). Then, records from all databases were merged and duplicates with identical coordinates for the same species were removed. We obtained almost 30 million (29,715,021) observations for 52,725 species. There were between 1 and 895,762 observations per species with a median of 196. The observations were then aggregated to the 30-arc second-pixel level to match the resolution of the predictor layers. Duplicate occurrences that fell within the same grid cell were removed and observations that fell off the pixel grid used for the predictors were removed (e.g. observations close to a coast that are aggregated to a pixel in the water). In subsequent steps, we consider the 24,140 species for which there were at least 20 spatially explicit observations after aggregation, which may be considered a reasonable minimum number of observations for modeling[58].

### Geographic range polygon

For each species, we constructed a geographic range that was used to select observations to be included in the model training, thus excluding observations of non-native or invasive species, and for model projection, to prevent predictions outside the native range. The same range was used for model projection for all considered climate scenarios. The range for each species was constructed based on the reported native countries from GlobalTreeSearch[39] and the location of the observations.

The reported native range consisted of the geometries of the countries in which the species is listed as native with a 1000 km buffer to compensate for potential gaps in the database. The particularly large buffer size was also chosen to allow species to spread under projected future climates; however, this will consequently expand the range beyond the native countries. The geometries for country boundaries were obtained from the Global Administrative Unit Layers (GAUL) dataset, implemented by FAO. Particularly when the reported native range contained large countries, it was important to exclude the areas in which no observations were found. For this purpose, we intersected the reported native range with a polygon constructed around the ecoregions[48] in which there were observations that had at least 3 other observations within 1000 km. For small ecoregions (bounding box less than 1000 km in width or length), the ecoregion with a 1000 km buffer was included in the polygon. For larger ecoregions, a 1000 km buffer around the part of the ecoregion that is within 200 km of an observation was included in the polygon. For computational efficiency, if there were more than 10,000 observations, these were spatially aggregated for the construction of the range.

### Training data

The training data consisted of the observations that fell within the considered range and pseudoabsences sampled uniformly at random within the considered range. For most species, the number of sampled pseudoabsences ($n_{PA,training}$) was 10 times the number of observations ($n_{obs,training}$). If there were less than 500 observations, we sampled 5000 pseudoabsences such that the environmental conditions of the region were well represented in the training data. The number of points in the training data was capped at 20,000 to ensure the model could run in Google Earth Engine without running out of memory. Therefore, if there were more than 1818 observations, we sampled fewer pseudoabsences, and if there were more than 10,000 observations we randomly sampled 10,000 observations and 10,000 pseudoabsences (Table 2).

**Table 2 | The number of observations and pseudoabsences included in the training data ($n_{obs,training}$ and $n_{PA,training}$) dependent on the number of observations after aggregation ($n_{obs}$)**

| $n_{obs}$ | $n_{obs,training}$ | $n_{PA,training}$ | $n_{total}$ |
|---|---|---|---|
| $n_{obs} >= 10{,}000$ | 10,000 | 10,000 | 20,000 |
| $1818 <= n_{obs} < 10{,}000$ | $n_{obs}$ | $20{,}000 - n_{obs}$ | 20,000 |
| $500 <= n_{obs} < 1818$ | $n_{obs}$ | $n_{obs} * 10$ | 5500 - 20,000 |
| $n_{obs} < 500$ | $n_{obs}$ | 5000 | 5000 - 5500 |

### Environmental variables and climate change scenarios

For the environmental niche model modeling, we selected nine environmental variables related to climate and soil conditions as predictive variables, based on preliminary variable importance analysis on a random subset of species using a random forests model. The climate variables consisted of average annual temperature, temperature seasonality, annual precipitation, precipitation seasonality, growing season length, and potential net primary production, and were obtained from CHELSA V2.1[94,95]. These factors represent basic resource requirements, metabolic modifiers or disturbance constraints to plant growth and survival. The soil variables consisted of soil pH, coarse fragment content, and silt content, and were obtained from SoilGrids[47]. We extracted all variables at a 30 arc-second resolution and projected them to the World Geodetic System 1984 (EPSG:4326) projection.

### Species distribution modeling

For each species, we fitted an ensemble model consisting of two random forests and two gradient tree boost classifiers, with different complexity levels with regard to model formulation[96]. The final distribution map was then achieved by taking the average over the predictions of the four individual models. We ran the model for 24,140 species for which at least 20 spatially explicit observations were available after aggregation.

To avoid overfitting when modeling rare species, we limited the number of predictor variables used in each model based on the number of observations for the corresponding species. We ensured that the number of observations available was at least 10 times the number of predictors used. For species that had less than 90 observations, the most important predictors for each species were selected based on the variable importance obtained when training a random forest model with all predictors. The number of selected predictors was determined by the number of occurrences in the training data (e.g., two predictors were selected for species with 20-29 occurrences). All nine predictors were used for species that had at least 90 observations.

The model output was probabilistic, and we used a 3-fold cross-validation with random fold assignment to determine the optimal threshold to transform the probabilistic output to binary, by assessing the threshold maximizing the true skill statistic (TSS). We evaluated our models by computing TSS, as well as the area under the ROC curve (AUC) during cross-validation.

Finally, the ensemble model was trained on the full training set and predictions were made on global maps of the covariates clipped to the considered range for each species. While soil variables were kept constant, we performed the predictions on historical averages of data from 1981-2010, referred to as current climate, and nine sets of variables for future climate scenarios: three time periods (2011-2040, 2041-2070, 2071-2100) and three different socio-economic pathways (SSP's), representing a sustainability scenario (SSP 1.26), a regional rivalry scenario (SSP 3.70), and a fossil fuel development scenario (SSP 5.85). We used five different global circulation models (GCMs) from the sixth coupled model intercomparison project (CMIP6): GFDL-ESM4,

UKESM1-0-LL, MPI-ESM1.2-HR, IPSL-CM6A-LR, and MRI-ESM2.0. They were bias-corrected[97] for the intersectoral-impact model inter-comparison project (ISIMIP)[98]. For computational efficiency, the distributions were computed with a single model considering the average values over the five models, rather than taking the average of the predictions of five models each considering one GCM.

For further analyses, based on model evaluation through cross-validation (Supplementary Fig. 1), we considered only the 10,590 species for which at least 90 spatially explicit observations were included in the training data. The distributions were further evaluated on independent presence-absence data from sPlot[40]. Data was available for 5939 of the 10,590 considered species and covered 47,479 plots across the globe. However, the comparison was computed only for the 3594 species that had at least 5 presences recorded in plots within the species' geographic range. For every species, a confusion matrix was constructed comparing the species' distribution model output to the survey data from plots within the species' geographic range, and the TSS, precision, and recall were computed. Finally, we compared our modeled distributions to maximum habitat suitability maps for 23 European species from the "Tree species distribution data and maps for Europe" report from the European Commission[41] by computing their overlap with intersection over union (IoU).

### Taxonomic and phylogenetic ordination

We extracted a global community matrix for the 10,590 considered tree species by extracting the predicted species distribution for the current climate at 100 km resolution using the equal area projection EPSG:6933[99], obtaining 12,548 sites. The distributions had to be sampled at a relatively low resolution due to computational feasibility of the ordinations. Distances between sites were computed with the Sorenson distance using the vegdist() function from the vegan R package[100] (2.6.4). The taxonomic ordination was computed as a 3-axis non-metric dimensionality scaling (NMDS) with the metaMDS() function with default parameters from the vegan R package[100] (2.6.4). We obtained a satisfactory stress value of 0.04. Each axis was mapped to the red, green, and blue values for visualizations. The phylogenetic ordination was computed with the evopcahellinger() function in the adiv R package[46] (2.2) using the same community matrix and a phylogenetic tree constructed with the phylo.maker() function in the V.PhyloMaker R package[44] (0.1.0). The first three axes of the ordination were retained. They explain 26.4%, 10.6% and 6.5% of the variation, respectively, and were also mapped to the red, green, and blue values.

The clustering analysis was performed using the sklearn.cluster.Kmeans() function and evaluated with the sklearn.metrics.silhouette_score() function from the scikit-learn python package[101] (v1.1.3). The 3-axes of each ordination were clustered together after the removal of outliers, defined as points that fall outside of the range [Q1-IQR, Q3 + IQR] where Q1 and Q3 are the first and third quartiles and IQR is the interquartile range (ie. Q3-Q1). Maps of the clustered ordinations were created with the number of clusters that generated the highest silhouette score. Redundancy analyses were performed on the outputs from both ordinations with the rda() function and variation partitioning analyses with the varpart() function, both from the vegan R package[100] (2.6.4).

A principal component analysis was applied to the environmental variables extracted at a 100 km resolution, to match the sites of the ordinations. We used the sklearn.decomposition.PCA() function from the scikit-learn python package[101] (v1.1.3). The two first principal components were used to represent the taxonomic and phylogenetic ordination results in a 2-dimensional representation of the environmental space.

### Occupancy range size

We obtained the species range sizes by computing the surface area covered by the estimated species distribution for the current climate

variables (1981-2010). Range sizes were computed for the full distributions and for the distributions constrained to forested areas, defined as areas with at least 10% tree cover as estimated using remote sensing data from the year 2000[29]. Although maps of tree cover corresponding to more recent years are available, we used the one from the year 2000 because it corresponds roughly to the period represented by the climatic data. Biome-level statistics were computed using the forest biomes from the RESOLVE classification[48] with species assigned to a biome if at least 20% of their estimated distribution fell within that biome. The global map of median range size was obtained by computing the median of the range sizes for all species that were predicted to be suitable in each 30-arc-second pixel.

### Species response to climate change
To assess the response of tree species distributions to climate change, we computed differences between the predicted distribution maps with climatic variables for 1981-2010 and for 2041-2070 with SSP 5.85. The most distant and extreme climate change scenario was selected such that differences would be maximized. However, as species-level predictions in the future have many uncertainties, trends were investigated instead on the ecoregion-level, using the ecoregions from forest biomes using the RESOLVE classification[48]. The fraction of species predicted to be gained and lost under climate change was computed for each ecoregion using the species predicted to be found within each ecoregion's boundaries for both sets of climatic variables. The median elevation shift and absolute latitude shift were computed for each ecoregion over the species predicted to be present in an ecoregion with both current and future climatic variables, by taking the shift in median latitude and elevation between current and future distributions for each species. Taxonomic and phylogenetic ordinations were computed on the ecoregion-level community matrix, considering both current and future compositions simultaneously. The same functions as for the global taxonomic and phylogenetic composition analysis were used. We obtained a stress value of 0.08 for the 3-dimensional NMDS, and the three first axes of the phylogenetic PCA explained 40%, 15%, and 7% of the variation. The change composition was computed as the Euclidean distance between scaled ordination values. Biome-level means and 95% confidence intervals were computed over the ecoregions assigned to each biome according to the RESOLVE ecoregion and biome classification[48].

### Reporting summary
Further information on research design is available in the Nature Portfolio Reporting Summary linked to this article.

## Data availability
The occurrence data used in this study are available from the online databases which are listed in Table 1. One unpublished dataset was used; it is available from the corresponding author upon request. The bioclimatic raster data used as used model covariates in this study are available from CHELSA 2.1[94,95]. The edaphic raster data used as used model covariates in this study are available from Soilgrids[47]. The reported native ranges data used in this study are available from GlobalTreeSearch[39]. The country boundary data from the FAO used in this study are available through the Google Earthengine data catalog (https://developers.google.com/earth-engine/datasets/catalog/FAO_GAUL_2015_level0). Plot data used as an independent validation data in this study area available from sPlot[40]. The biome and ecoregion database used in this study are available through the Google Earthengine data catalog (https://developers.google.com/earth-engine/datasets/catalog/RESOLVE_ECOREGIONS_2017). The tree cover map used in this study are available through the Google Earthengine data catalog (https://developers.google.com/earth-engine/datasets/catalog/UMD_hansen_global_forest_change_2023_v1_11). Source Data for Figs. 1, 2 and 3 are provided with this paper as Source Data files. The raster

data of the modeled tree species' distributions generated in this study have been deposited in Zenodo: https://doi.org/10.5281/zenodo.10911892[102]. Source data are provided with this paper.

## Code availability
The code used for both the modeling pipeline and the downstream analyses can be found in the following GitHub repository: https://github.com/ninavantiel/tree_sdms[103].

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

## Acknowledgements

This project was partly financed by the Internal WSL grant Treemap (LP), the Internal WSL grant ClimEx (D.N.K. and N.E.Z.), the BiodivERsA project FeedBaCks with the national funder Swiss National Science Foundation (20BD21_193907) (D.N.K. and N.E.Z.), BiodivERsA project Futureweb with the national funder Swiss National Science Foundation (20BD21_184131) (DNK), the deepHSM project with the national funder Swiss National Science Foundation (204057) (N.v.T. and D.T.). N.v.T., J.v.d.H and T.W.C. acknowledge funding from DOB Ecology. N.v.T. acknowledges funding from the Restor Foundation. We thank to David Boehm for data cleaning and merging as part of his Swiss civil service. We thank the following organizations for making their data freely available: BIEN, BIOMASS, Chelsa, CONIFER, DRYFLOR, GBIF, GFBI, GlobalTreeSearch, IDIGBIO, INDIABIODIVERSITY, PNG, PREDCITS, RAINBIO, sPlot, SoilGrids.

## Author contributions

N.v.T., T.W.C. and L.P. conceived and designed the analysis. N.v.T., F.F., P.B., J.v.d.H., D.N.K., C.M.C., L.L. and N.E.Z. contributed analysis tools. N.v.T. performed the analysis. N.v.T. and L.P. wrote the original draft of

the paper with assistance from T.W.C., D.T. and N.E.Z. All authors contributed to revisions and editing of the final version.

## Competing interests

The authors declare no competing interests.
