## [Peer Review File · Nature Communications]

Regional uniqueness of tree species composition and response to forest loss and climate changeREVIEWER COMMENTS

Reviewer #1 (Remarks to the Author):

I found the paper well-written and intriguing. The extent of analysis is huge, and a Nat Comms paper is a short space to give such complexity appropriate attention. Still, the authors do manage to pull out meaningful results. I am generally not a fan of very 'mega' papers like this, but the authors do seem to have been pretty careful in their choice of methods and acknowledge limitations in the main text. Thus, overall, I actually only have fairly minor suggestions, detailed below.

How is their tree species mapping different from previous efforts? I would assume they think theirs is better, but this needs to be explained. For example, see:

Serra-Diaz, J. M., Enquist, B. J., Maitner, B., Merow, C. & Svenning, J. C. Big data of tree species distributions: how big and how good? *For. Ecosyst.* 4, 0–12 (2017).

One of the main results emphasised is the 'high uniqueness' of composition in any given place on the globe, but this is not backed up by any analysis. This statement needs to be justified somehow.

The authors could have validated their modelling for regions where tree species have their ranges mapped. That would be a good proof of concept.

The authors use 10% tree cover as their threshold for forest. What about trees in savannas? Or the work showing the large number of trees occurring in drylands (e.g., Reiner et al. 2023 Nat Comms, Brandt et al. 2020. Nature)

Can they give us more info on this "which limits the number of false negatives while tolerating some false positives"? That is rather vague. What is 'some false positives'? Can this be estimated? Also, is this mis-phrased: "false positives are not necessarily problematic when comparing presence-absence data to habitat suitability maps, as the absence of a species does not mean the habitat is not suitable" I would have expected that sentence to be about false negatives.

Figure 1 is confusing. Why not show the actual result of the ordinations, rather than colouring points based on position in 3 dimensional space. Environmental vectors can be mapped onto the ordinations. I think it is important that the reader sees these ordinations in the main text to know how good they are.

Lines 209-210 - small ranges in tropics don't necessarily reflect ecological specialism, but could just be product of non-equilibrium dynamics (sensu Hubbell or Ricklefs)

Give species authority when first mentioning it.

Results section actually includes lots of discussion of results (e.g. lines 209-214, 254-255), but not referencing the literature (e.g., lines 172-174... this is not a novel result... see work by Qian and Ricklefs and others). Discussion either needs to be moved to discussion or appropriate references should be placed in that section.

Why use 2000 as forest cover year? Lots of deforestation since then and data are out there for more recent years.

Lines 229-230 - I would advocate using the IUCN ecosystem typology. Those ecosystems (and biomes) are functional and probably make more sense to use.

The elevational range shifts in the results need more explanation.

Lines 265-266 - Given the quality of globally available soils data, it is not surprising that edaphic factors do not come out as that important. This should be acknowledged.

Line 338 - Although their figure 2d suggests that the majority of species at tropical latitudes have very small ranges

Line 415 - why these particular soil variables? Please justify.

Line 459 - what was the stress value of this NMDS? It seems unlikely that it would be below

an acceptable threshold for a community composition dataset spanning the globe.

Line 464 - how much variation explained by these three axes?

Reviewer #2 (Remarks to the Author):

In this manuscript, entitled "Regional uniqueness of tree species composition and response to forest loss and climate change", the authors investigated the species composition using 10590 tree species globally, and tree species range dynamics in both historical and future contexts. While the subject matter is captivating, there are some major concerns about the methodology and overall logic presented in the current version.

The manuscript should include detailed step-by-step instructions for the main analyses conducted, making it easier for readers to follow and repeat the process. The current version of the manuscript lacks comprehensive details in several areas, such as the data cleaning process. Specifically, the authors mentioned removing duplicates from occurrences collected from 13 databases, but this section requires further elaboration. As the accuracy and completeness of the compiled data are fundamental to the study, a more focused and in-depth explanation of the data cleaning process is essential. In addition, to strengthen the validity of the compiled data, it would be ideal to provide additional information about the validation process. While the authors briefly mentioned validation using sPlot data in Line 436, this section should be expanded to instill confidence in the dataset used. Some specific comments to the method are listed below.

The link between the main results concerning the uniqueness of species composition and species range changes seems rather weak now. While phylogenetic diversity was utilized in the first section, its role in subsequent analyses should be clarified. Exploring how species composition might change in the future would be an interesting addition, potentially offering deeper insights into the implications of climate change on tree species distributions.

Specific comments:

Line 53-57, did the authors indicate those factors drive all the regional turnover globally?

Probably not.

Lines 60-63, it is not completely correct, two global studies on tree species richness have been published during the last years: Liang, J., Gamarra, J.G.P., Picard, N. et al. Co-limitation towards lower latitudes shapes global forest diversity gradients. *Nat Ecol Evol* 6, 1423–1437 (2022). & Guo, W.Y. et al, Paleoclimate and current climate collectively shape the phylogenetic and functional diversity of trees worldwide. *bioRxiv* doi:10.1101/2020.06.02.128975 (2020).

Line 64, it is not clear why phylogenetic composition is important here. Probably needs some introductory content previously.

Line 73-74, again, see the two papers for Lines 60-63. And a resolution of 100 km used here could be a bit coarse for some experts.

Line 85, could be helpful to specify what “climate-smart” refers to.

Lines 57-91, I agree that these are important. However, after reading the manuscript, it seems this study did not overcome those limitations, or the algorithms used did not seem novel to many SDM users.

Lines 170-172, maybe I missed something, but I did not see this pattern from Fig, S4.

Line 265, were any results here related to this statement (historical factors)?

Lines 272- 285, it is true these techniques are important to macro-scale ecology studies; however, I do not see this paragraph is relevant here, particularly as the second paragraph of the discussion.

Lines 383 and on, why this 1000 km was selected? It is not a short distance and over the size of many European countries, thus, could cover nonnative countries.

Lines 427-430, it is a bit confused and how to be sure which variables are important to specific species?

Reviewer #3 (Remarks to the Author):

The manuscript is interesting and have great potential, I like the idea and I also mostly like the multiple methodology used to develop the research, however there are some issues that need attention.

L28-29 “Nevertheless, tropical moist and boreal forest biomes still harbor extremely large29

ranged tree species.” Well this is not necessarily entirely true because these are the two most extreme forest biomes, the former well known to be stable since a long period of time, the latter composed by few species that have large distributional ranges. I think this need careful interpretation.

L30 “to differ IN” I think in should not be capitalised.

L96 “for which we had sufficient occurrence data” how was this assessed?

L113-114 “combining geographic range polygons based on reported native countries and species distribution modelling” I have some concern about the use of distribution at country level because each country have very different size and thus this cannot really be considered a measurement unit. The authors explained this in the method section but this should be better highlighted here as well.

L115 “at a 30-arc second resolution” this is my main concern, I strongly suggest to change the projection of the study into an equal-area projection because this is the way how global spatial biodiversity assessments should be made, species-presence-absence and associated processes are scale and spatial dependent.

L117-118 “with at least 20 spatially explicit observations and available reported native ranges from GlobalTreeSearch” why 20 observations is considered an adequate number? And how much should they spatially separated?

L119, I think the authors should clearly separate the choice of selection the species to be modelled: from one side based on biological basis (e.g. sufficient number of observation representative of the natural range) and on technical basis (e.g., sufficient number of observation to run reliable models).

L144-146 I would consider the fire season length instead of too narrow environmental variables like soil properties.

Fig 1, I am really surprised that both taxonomic and phylogenetic ordinations are so similar, is this related to the methodology used (e.g. evopcahellinger)?

Fig2-Fig3 how have been the biomes identified? Which classification have been used?

Dear reviewers,

We would like to thank you for the opportunity to submit a revised version of our manuscript “Regional uniqueness of tree species composition and response to forest loss and climate change” for publication in *Nature Communications*. We thank you for your time reviewing our manuscript and your constructive comments that allowed us to greatly improve our study. We provide a point-by-point response to each of the reviewer’s comments below. All modifications to the manuscript, including slight changes in the wording of certain sentences, have been marked in blue. Finally, we added data and code availability statements to our manuscript, including a link to the code necessary to replicate our study.

Reviewer #1 (Remarks to the Author):

I found the paper well-written and intriguing. The extent of analysis is huge, and a Nat Comms paper is a short space to give such complexity appropriate attention. Still, the authors do manage to pull out meaningful results. I am generally not a fan of very 'mega' papers like this, but the authors do seem to have been pretty careful in their choice of methods and acknowledge limitations in the main text. Thus, overall, I actually only have fairly minor suggestions, detailed below.

Response: We would like to thank the reviewer for the positive evaluation of our work and for the pertinent comments.

How is their tree species mapping different from previous efforts? I would assume they think theirs is better, but this needs to be explained. For example, see: Serra-Diaz, J. M., Enquist, B. J., Maitner, B., Merow, C. & Svenning, J. C. Big data of tree species distributions: how big and how good? *For. Ecosyst.* 4, 0–12 (2017).

Response: The main technical innovation provided for our mapping pipeline was the implementation of a species distribution modeling algorithm in a cloud-computing platform. Additionally, our study benefits from the large amounts of publicly available occurrence data and our efforts to compile multiple databases. The size of our dataset is on par with that of the Serra-Diaz et al. paper. However, while the paper by Serra-Diaz et al. presents an impressive analysis of the geographical coverage of occurrence data globally, it does not include any species distribution modeling as in our study. Here, the combination of large datasets and our cloud-based environmental niche algorithm allowed us to generate continental-wide maps at a higher resolution than previously done, of 30 arc seconds, for a large number of species and with high precision thanks to the high number of occurrence records. We modified lines 113-114 to clarify these two aspects. Furthermore, as underlined in the manuscript in lines 145-147, our approach combines species distribution modeling and geographic range limits, which was not the case with previous large-scale modeling efforts.

One of the main results emphasised is the 'high uniqueness' of composition in any given place on the globe, but this is not backed up by any analysis. This statement needs to be justified somehow.

Response: Thank you for this very relevant comment. We agree that our paper lacked some analyses to support the result of near-unique tree species composition across the Earth. We performed some additional analyses which are illustrated in figures in the supplementary material and are included in the results section.

Figure S5 shows the distributions of the ordination axes for both the taxonomic and phylogenetic ordinations in one- and two-dimensions. Figure S6 illustrates our revisited clustering analysis of the two ordinations: a plot of the silhouette coefficient that measures cluster quality against the number of clusters, and maps of the clustered taxonomic and phylogenetic ordinations using the number of clusters that maximized the score. The analysis is explained in the methods section in lines 507-513.

The lack of structure in the distributions combined with the lack of well-defined, fine-grained clusters indicates the uniqueness of tree species composition across the globe. We note that the taxonomic ordination showed smooth gradients with the best clustering generating only two clusters, which is not particularly informative as they retain a high level of intra-cluster variance. The phylogenetic ordination contains more structure - the most striking example of this is a peak found in the phylogenetic ordination around 0.25, 0.0, and -0.35 on each axis respectively (Figure S5f). This corresponds to an area of very homogeneous phylogenetic composition spanning Northern Canada and Alaska (Figure 1d) and one of the five clusters obtained for this ordination. We included these results in lines 176-183. These elements were also included in the discussion in lines 316-318.

The authors could have validated their modelling for regions where tree species have their ranges mapped. That would be a good proof of concept.

Response: We agree that this would provide a good validation. Hence, we have included a comparison of our modeled distributions to the maximum habitat suitability (MHS) maps from the "Tree species distribution data and maps for Europe" report from the European Commission (<https://data.europa.eu/doi/10.2760/489485>) in Figure S4, in the results section in lines 141-144, and the methods section in lines 488-491. We computed the intersection over union (IoU) of the areas covered by both maps for 23 species for which the data was available. While there are some considerable differences between both maps, we note that vast areas of agreement are found and that the discrepancies are generally due to our models being more conservative.

The authors use 10% tree cover as their threshold for forest. What about trees in savannas? Or the work showing the large number of trees occurring in drylands (e.g., Reiner et al. 2023 Nat Comms, Brandt et al. 2020. Nature)

Response: We agree that constraining species range to areas with at least 10% tree cover excludes parts of species' ranges that may occur outside of forests. This may lead to underestimated realized ranges of some tree species. However, we still consider that the range reduction is indicative of the extent of historical forest loss. We agree it is important to acknowledge this limitation and have added it in the discussion in lines 331-333.

Can they give us more info on this "which limits the number of false negatives while tolerating some false positives"? That is rather vague. What is 'some false positives'? Can this be estimated?

Response: We agree that this was a rather vague formulation. We reformulated this sentence to indicate that the most frequent type of errors made by our models were false positives (predictions of suitable habitat where the species has not been recorded) and included the false positive and false negative rates in lines 135-137.

Also, is this mis-phrased: "false positives are not necessarily problematic when comparing presence-absence data to habitat suitability maps, as the absence of a

species does not mean the habitat is not suitable" I would have expected that sentence to be about false negatives.

Response: We agree that this sentence was not very clear. However, it is intended to be about false positives, which are locations that the model has classified as suitable although no occurrence was recorded there (ie. falsely classified as positive/suitable). As we use presence-only data, the species may be present but not recorded. Alternatively, the habitat may be suitable with regard to the variables taken into account in the model even if the species is not found there due to other factors, such as biotic interactions, human influence, or stochasticity. We rephrased this sentence to clarify it in lines 137-140.

Figure 1 is confusing. Why not show the actual result of the ordinations, rather than colouring points based on position in 3 dimensional space. Environmental vectors can be mapped onto the ordinations. I think it is important that the reader sees these ordinations in the main text to know how good they are.

Response: We agree that our visualization of the ordinations in environmental and geographic space may be unconventional. We also appreciate and agree with the importance of visualizing the ordinations in the ordination space. Therefore, we added the visualizations in ordination space with 2-dimensional plots in the supplementary material in Figure S4, with the same coloring scheme as used in Figure 1. This figure is referenced in the main results in lines 168 and 170. Nonetheless, we prefer to keep the ordinations mapped to colors in a 3-dimensional space, as this allows us to create maps showing more than one ordination axis.

Moreover, as these ordinations are in taxonomic and phylogenetic space, respectively, we cannot map the environmental vectors in the ordination space in Figure S4. Environmental vectors are illustrated in Figure 1 a,c, where we see that the same environmental conditions can be associated with different ordination values.

Lines 209-210 - small ranges in tropics don't necessarily reflect ecological specialism, but could just be product of non-equilibrium dynamics (sensu Hubbell or Ricklefs)

Response: Thank you for pointing this out. As per your comment below that some discussion elements were present in the results, we removed this point from the results and included a sentence about it in the discussion in lines 360-361.

Give species authority when first mentioning it.

Response: We added that we used the species names from the GlobalTreeSearch checklist in the methods section in lines 386-388.

Results section actually includes lots of discussion of results (e.g. lines 209-214, 254-255), but not referencing the literature (e.g., lines 172-174... this is not a novel result... see work by Qian and Ricklefs and others). Discussion either needs to be moved to discussion or appropriate references should be placed in that section.

Response: Thank you for pointing this out. The discussion elements that were in the results concerning Figure 1 were already included in the discussion. Therefore, we simply removed them from the results section. The elements that were in the results for Figures 2 and 3 were moved to the discussion in lines 366-372.

Why use 2000 as forest cover year? Lots of deforestation since then and data are out there for more recent years.

Response: We acknowledge that we had not specified the reasons for this choice. We chose the tree cover from the year 2000 to temporally match the climatic covariates we used for current species distributions (1981-2010). This was added in the methods section in lines 526-528.

Lines 229-230 - I would advocate using the IUCN ecosystem typology. Those ecosystems (and biomes) are functional and probably make more sense to use.

Response: We used the biomes and ecoregions from RESOLVE documented in Dinerstein, E. *et al.* (2017). These are commonly used in ecological research to describe large-scale patterns in the distribution of ecosystems. They were derived from a qualitative effort informed by expert opinion and reflect the distributions of regional biotas based on species distributions. We consider such ecoregions, ie. regional ecosystems, to be adequate for this type of analysis.

Although the IUCN ecosystem typology is interesting as it considers functional groups, it allows the same location to be assigned to multiple functional groups simultaneously and regional polygons are not provided. On the other hand, the RESOLVE ecoregions are mutually exclusive, i.e., a given location is only associated with one ecoregion and one biome. We therefore consider this to be better suited to the setup proposed for our study.

The elevational range shifts in the results need more explanation.

Response: We added some explanations and references about the elevational range shifts in lines 257-259.

Lines 265-266 - Given the quality of globally available soils data, it is not surprising that edaphic factors do not come out as that important. This should be acknowledged.

Response: Thank you for this pertinent comment. We had actually only quantified the explained variance of the ordinations by climatic variables but your comment pushed us to include a more in-depth analysis. We, therefore, included a redundancy analysis of the ordinations with all model covariates (climate and soil variables) and a variation partition analysis to quantify the variation explained by edaphic or climatic factors, or by both sets simultaneously. An illustration of the variation partition analysis results can be found in the supplementary material in Figure S7. These analyses confirmed that edaphic factors were significantly less important than climatic factors. We hypothesize that this may be due to scale dependency or to the fact that the soil data was generated by models that rely on climate-related variables. These findings were added to the results in lines 183-190.

Line 338 - Although their figure 2d suggests that the majority of species at tropical latitudes have very small ranges

Response: Thank you for pointing out this inconsistency. We rephrased this to specify that the species with large ranges are found specifically in the tropical moist broadleaf forest biome and that they coexist with many species with small ranges. These modifications can be found in lines 354-357.

Line 415 - why these particular soil variables? Please justify.

Response: We selected these specific climatic and soil variables based on preliminary variable importance analysis on a subset of species and using a random forest model. We added this in the methods section in lines 438-439.

Line 459 - what was the stress value of this NMDS? It seems unlikely that it would be below an acceptable threshold for a community composition dataset spanning the globe.

Response: We obtained a stress value of 0.04 which we consider satisfactory, as stress values under 0.05 are generally considered a very good representation in reduced dimensions. We added this in line 500.

Line 464 - how much variation explained by these three axes?

Response: The three first axes of the evoPCA explained 28.6%, 9.8% and 6.3%. We included this in line 505.

Reviewer #2 (Remarks to the Author):

In this manuscript, entitled " Regional uniqueness of tree species composition and response to forest loss and climate change", the authors investigated the species composition using 10590 tree species globally, and tree species range dynamics in both historical and future contexts. While the subject matter is captivating, there are some major concerns about the methodology and overall logic presented in the current version.

Response: We would like to thank the reviewer for the positive evaluation and for the interesting points raised below, which we have addressed in the revised manuscript. We hope our answers below will solve the reviewer's concerns.

The manuscript should include detailed step-by-step instructions for the main analyses conducted, making it easier for readers to follow and repeat the process. The current version of the manuscript lacks comprehensive details in several areas, such as the data cleaning process. Specifically, the authors mentioned removing duplicates from occurrences collected from 13 databases, but this section requires further elaboration. As the accuracy and completeness of the compiled data are fundamental to the study, a more focused and in-depth explanation of the data cleaning process is essential.

In addition, to strengthen the validity of the compiled data, it would be ideal to provide additional information about the validation process. While the authors briefly mentioned validation using sPlot data in Line 436, this section should be expanded to instill confidence in the dataset used.

Response: We agree that details should be available for reproducibility. We therefore added details and explanations in the methods section. Details about data cleaning were added in lines 388-397. The source of the geometries used to construct the geographic range polygons was added in lines 411-412. Explanations of the validation process with the sPlot data were added in lines 482-488. Specifications about the taxonomic ordination were added in lines 496-500, and about the clustering, redundancy, and variation partitioning analysis in lines 507-515. Explanations for the occupancy range size analysis and the calculation of species responses to climate change that we conducted to obtain Figures 2 and 3 were

added in lines 521-553. Furthermore, the code necessary to reproduce our study is available in a public repository which is linked in the code availability section in line 559.

The link between the main results concerning the uniqueness of species composition and species range changes seems rather weak now. While phylogenetic diversity was utilized in the first section, its role in subsequent analyses should be clarified. Exploring how species composition might change in the future would be an interesting addition, potentially offering deeper insights into the implications of climate change on tree species distributions.

Response: Thank you for this insightful comment. We had originally performed each analysis separately and had not linked the species composition results to those concerning the effect of climate change. This comment encouraged us to compute the ecoregion-level changes in taxonomic and phylogenetic composition considering the distributions estimated for current climate variables and those estimated under climate change. These results can be found in Figure 3, with the adapted legend in lines 227-235. The results were included in the paragraph found in lines 263-271. The relevant details were added in the methods in lines 546-551. We note that the other results concerning the effect of climate change were recomputed in the process and some slight adjustments were made in the results in lines 246-267.

Line 53-57, did the authors indicate those factors drive all the regional turnover globally? Probably not.

Response: Our sentence here was not phrased clearly. We reformulated it to indicate that variations in freezing frequencies or moisture are an example of factors that drive turnover in the Americas. The revised sentence can be found in lines 58-59.

Lines 60-63, it is not completely correct, two global studies on tree species richness have been published during the last years: Liang, J., Gamarra, J.G.P., Picard, N. et al. Co-limitation towards lower latitudes shapes global forest diversity gradients. *Nat Ecol Evol* 6, 1423–1437 (2022). & Guo, W.Y. et al, Paleoclimate and current climate collectively shape the phylogenetic and functional diversity of trees worldwide. *bioRxiv* doi:10.1101/2020.06.02.128975 (2020).

Response: Thank you for pointing this out. We included the references to these relevant studies in our manuscript. However, we note that both studies base their diversity estimates on occurrence data rather than on species distribution modeling outputs. This may underestimate diversity in regions where we do not have complete observational data. We accordingly adapted this section of the manuscript in lines 62-67.

Line 64, it is not clear why phylogenetic composition is important here. Probably needs some introductory content previously.

Response: We agree that some introductory content was lacking. We added this earlier in the paragraph in lines 50-52.

Line 73-74, again, see the two papers for Lines 60-63. And a resolution of 100 km used here could be a bit coarse for some experts.

Response: We agree that 100 km resolution of would be considered coarse. However, our distributions were estimated at a resolution of 30 arc seconds (~1 km). The species

distributions were only aggregated to 3,000 arc seconds (~100 km) to compute the composition ordinations for Figure 1. We added the resolution of the distributions in line 98 to clarify this. Moreover, the Liang et al. paper directly estimates species richness and does not estimate species' ranges for a large number of species globally. The study by Guo et al. does, on the other hand, obtain species' ranges but with alphaHulls around observations only, rather than combining species distribution modeling with geographic range constraints as we did. Moreover, they use a resolution of about 100 km. Accordingly, we added a reference to the Guo et al. paper as an example of a coarser-resolution species range mapping in line 79.

Line 85, could be helpful to specify what "climate-smart" refers to.

Response: We agree the term "climate-smart" may not be clear and have replaced it with "climate change-resilient" in line 90.

Lines 87-91, I agree that these are important. However, after reading the manuscript, it seems this study did not overcome those limitations, or the algorithms used did not seem novel to many SDM users.

Response: We agree that our approach does not overcome these limitations. Nevertheless, we consider it important to mention them. We reformulated the sentence in lines 94-96 to clarify that we did not overcome these limitations, but that approaches such as ours may still be relevant.

Lines 170-172, maybe I missed something, but I did not see this pattern from Fig, S4. "In contrast, the subtropical environments surrounding the Indian Ocean show much higher similarities in phylogenetic than taxonomic composition."

Response: Following a comment from Reviewer 1, we modified our clustering analysis, with a more thorough assessment of the optimal number of clusters. We now find 5 clusters in the phylogenetic composition and the areas surrounding the Indian Ocean are clustered together. We consider them more similar in phylogenetic than taxonomic composition based on the colours on the maps in Figure 1. Furthermore, even though they are in the same cluster in the taxonomic composition clustering, the 2 clusters we obtain here still contain a large amount of variation, whereas the phylogenetic composition clusters are of better quality with a larger silhouette score (Figure S6). Finally, following a comment by Reviewer 1, the lines referred to were removed from the results, as it was already present and more appropriate in the discussion (lines 307-308).

Line 265, were any results here related to this statement (historical factors)?

Response: Our study does not quantify the effect of historical factors on the tree species composition. We rephrased this sentence to clarify that our results quantified the associations with ecological factors and we suggest that residual variance may be attributed to other factors such as historical factors. This can be found in line 279.

Lines 272- 285, it is true these techniques are important to macro-scale ecology studies; however, I do not see this paragraph is relevant here, particularly as the second paragraph of the discussion.

Response: Thank you for your comment. We consider that the main technical innovation that we provided for our mapping pipeline was the implementation of a species distribution modeling pipeline in a cloud-computing platform. This allowed us to generate continental-

wide maps at a high resolution of 30 arc seconds, which, in turn, allowed us to study entire species ranges and range shifts along latitude and elevation with sufficient precision. We, therefore, consider that highlighting the technical aspects and the datasets that allowed us to carry out this study is relevant to the discussion. Nonetheless, we shortened it slightly as we appreciate that these elements may not be of the main interest to some readers.

Lines 383 and on, why this 1000 km was selected? It is not a short distance and over the size of many European countries, thus, could cover nonnative countries.

Response: We agree that 1,000 km is a large buffer size. On one hand, it was selected to compensate for potential gaps in the native country dataset, as the lack of data in many regions due to sampling biases may lead to incomplete native-country datasets. On the other hand, as we wanted to use the same range polygon when predicting distributions with current and future climatic variables, we decided to add such a large buffer so that the range polygon would be large enough to allow the species distribution to shift under climate change. We acknowledge that this comes with the limitation that the range will therefore cover nonnative countries. We added explanations in the methods section in lines 409-411.

Lines 427-430, it is a bit confused and how to be sure which variables are important to specific species?

Response: We agree that this section was confusing. For species with fewer than 90 observations, we trained a random forest model with all predictors to obtain the variable importance for each predictor. Then, the top predictors were selected such that there were at least 10 observations per predictor (eg. two predictors were selected for species with 20-29 occurrence records) and the final model was trained with those predictors. We clarified this section in lines 456-461.

Reviewer #3 (Remarks to the Author):

The manuscript is interesting and have great potential, I like the idea and I also mostly like the multiple methodology used to develop the research, however there are some issues that need attention.

Response: We would like to thank the reviewer for the encouraging evaluation of our work. We hope we have addressed all issues in the revised version and our answers below.

L28-29 “Nevertheless, tropical moist and boreal forest biomes still harbor extremely large29 ranged tree species.” Well this is not necessarily entirely true because these are the two most extreme forest biomes, the former well known to be stable since a long period of time, the latter composed by few species that have large distributional ranges. I think this need careful interpretation.

Response: We modified the sentence to acknowledge that the tropical moist and boreal forest biomes have significant differences in richness and composition, while still conveying our results that these biomes have the lowest levels of range restriction across all other forest biomes and host large-ranged tree species. It can be found in lines 28-30. Furthermore, we would like to note that the results section contains a more extensive explanation of the differences between these two biomes in lines 216-224.

L30 “to differ IN” I think in should not be capitalised.

Response: Thank you for pointing this out. It was corrected in line 31.

L96 “for which we had sufficient occurrence data” how was this assessed?

Response: We agree that this is rather vague. We selected species with more than 90 occurrence records as this was found to train models with adequate predictive performance. This was clarified in lines 102-103. We also clarified this in the results section in line 128.

L113-114 “combining geographic range polygons based on reported native countries and species distribution modelling” I have some concern about the use of distribution at country level because each country have very different size and thus this cannot really be considered a measurement unit. The authors explained this in the method section but this should be better highlighted here as well.

Response: We agree that country-level native ranges are not ideal due to the variation in country sizes. However, to our knowledge, these are the most detailed global data that are currently available. Nonetheless, the ranges also depend on the location of occurrences within the reported native range and they represent only a coarse estimate of species’ native ranges. We added these details in lines 122-125.

L115 “at a 30-arc second resolution” this is my main concern, I strongly suggest to change the projection of the study into an equal-area projection because this is the way how global spatial biodiversity assessments should be made, species-presence-absence and associated processes are scale and spatial dependent.

Response: We agree that scale-dependency is an important issue when conducting spatial biodiversity analyses, but we consider it less pressing when modeling habitat suitability of individual species. In this case, scale dependency would boil down to the question of whether the environmental predictors systematically differ with area covered by a pixel. The Arctic tree line extends to about 70 °N, and therefore the area covered by pixels of our 30-arc grids differ by a factor of three, roughly between 342,000 m² and 1 km². Across these scales, we do not see an issue for the environmental covariates used here. Reprojecting these layers into equal area projections, on the other hand, would introduce errors into the model covariates which would propagate to the results. Nevertheless, although our modeling pipeline uses a 30-arc second resolution, the actual area of each pixel was taken into account for all relevant downstream analyses (range sizes in Figure 2 and latitude and elevation shifts in Figure 3).

L117-118 “with at least 20 spatially explicit observations and available reported native ranges from GlobalTreeSearch” why 20 observations is considered an adequate number?

Response: While the choice of the number of minimum observations to construct a model remains to some extent arbitrary, other studies have also used 20 as a minimum. We added a reference to one of these in line 396-397 (Serra-Diaz et al. 2018). More importantly, this is only taken as an initial minimum number, which we then increase to 90 following the results in Figure S1, which is the relevant number for the downstream analyses. We clarified this in the methods in lines 480-482.

And how much should they spatially separated?

Response: The observations are aggregated at 30-arc seconds, corresponding to the resolution of the model covariates. This means they are spatially separated by 30-arc seconds which corresponds to about 1 km at the equator. We clarified this in line 122.

L119, I think the authors should clearly separate the choice of selection the species to be modelled: from one side based on biological basis (e.g. sufficient number of observation representative of the natural range) and on technical basis (e.g., sufficient number of observation to run reliable models).

Response: Thank you for this comment. The criteria for the selection of species was made on a technical basis as we selected the number of observations with which the model performance is expected to be sufficient. We rephrased this in line 128 to clarify this.

L144-146 I would consider the fire season length instead of too narrow environmental variables like soil properties.

Response: We agree that variables that characterize fire season length or fire frequency and severity are likely to be important. However, as we used the same model covariates for all species, we selected variables of general ecological relevance across all biomes. Furthermore, the soil variables that we included can capture critical insights about the nutrient limitations that influence species ranges. We added a sentence in the discussion to address the limitations associated with the choice of model covariates in lines 295-297.

Fig 1, I am really surprised that both taxonomic and phylogenetic ordinations are so similar, is this related to the methodology used (e.g. evopcahellinger)?

Response: The two ordinations look quite similar as we chose the mapping of the ordination axes to red, green, and blue such that the colors would match as best as possible between both ordinations. However, the methodologies used for both ordinations are quite different, as explained below.

For the taxonomic ordination, we used NMDS which uses rank order of the dissimilarity between sites. Here, we used the Sorenson index (which is equivalent to the Bray-Curtis index for presence/absence data) to quantify the turnover in species composition between sites. On the other hand, for the phylogenetic ordination, we used an evolutionary PCA based on Hellinger distance, where the branch lengths of a phylogenetic tree, or evolutionary units, are used as the basic entities instead of species. This inclusion of evolutionary units in this approach makes it distinct from the approach used for the taxonomic ordination.

Fig2-Fig3 how have the biomes been identified? Which classification have been used?

Response: We used the RESOLVE biome classification. The appropriate reference (Dinerstein et al. 2017) was added in lines 242, 529 and 539.

References

- Dinerstein, E. *et al.* An Ecoregion-Based Approach to Protecting Half the Terrestrial Realm. *BioScience* **67**, 534–545 (2017)
- Thuiller, W. *et al.* Consequences of climate change on the tree of life in Europe. *Nature* **470**, 531-534 (2011).
- Serra-Diaz, J. M., Enquist, B. J., Maitner, B., Merow, C. & Svenning, J.-C. Big data of tree species distributions: how big and how good? *For. Ecosyst.* **4**, 30 (2018).

REVIEWER COMMENTS

Reviewer #1 (Remarks to the Author):

Overall, I think the authors have done a sufficient job with their revision. Please find below a few more minor comments.

Lines 208-210: "The relative decrease in range size reflects how species ranges are restricted by historical forest cover loss and we find a median range reduction of 26.7% across all considered species"

>there are lots of things that may prevent species from occupying their fundamental niche. I don't think this can be definitively attributed to forest loss. A more cautious statement here would be appropriate. I note that they are cautious in the discussion.

Lines 279: Or to historical and regional factors sensu Ricklefs.

It would be interesting to know the number of species modelled per biome.

Lines 306-310: "For instance, the phylogenetic similarity of India, subtropical Africa, and Australia highlight the signal of their past connection before their segregation caused by plate tectonics⁵⁷, while the similarity between temperate moist forests in North America and China is likely to indicate past connectivity of this biome followed by evolutionary conservatism⁵⁹."

>The authors are ignoring the increasing body of work showing lots of dispersal among continents since they separated. Most plant families are younger than continental splits. See classic papers by Susanne Renner and others (incl more recent one by Gagnon et al. in *New Phytologist*). Sorry, don't have internet while writing this review to look up citations!

The authors state in response doc, "Moreover, as these ordinations are in taxonomic and phylogenetic space, respectively, we cannot map the environmental vectors in the ordination space in Figure S4." This can readily be done with `ordifit` function in `vegan` package in R. This is more for their info. I am happy (enough) with the way the ordinations are now presented.

Lines 355-356: SE Asian forests have higher species richness than the Congo, so authors might consider rephrasing this sentence.

Reviewer #2 (Remarks to the Author):

I appreciated the revision which addressed my previous comments well. The major concern I have is about the species being able to include. As the authors said, they only kept species having more than 90 records for the further analyses. It indicated that those species are largely widely distributed. In contrast, lots of small-ranged species exist in the tropics (Gatti et al., 2022 PNAS; Guo et al., 2020 Preprint & 2022 PNAS; Liang et al., 2022 NEE), as the authors also mentioned in Line 357. The exclusion of those range-limited species in tropics could cause certain bias in the results, particular Fig.2a, which revealed that tropical moist forests hold lots of large range species. This finding is not incorrect, but could miss another peak in the small range size end. And this missing peak could be more vulnerable to future climate change and/or tree loss, that is, affecting the results of Fig. 2b & 3.

This reminded me another potential drawback about the SDM compared to other methods to obtain species' distribution, even though the reviewers thought it the "main technical innovation" of the study. I admired how the authors can manage to get the SDM models done, however, given its requirement of high-quality and large-quantity of occurrences, the majority of species will be excluded, just like the case of this study.

One minor point is the validation using 23 species. From Fig. S4 and also the relevant text, it seems the ranges were not well matched. The authors wrote their estimated ranges tended to be conserved. How big this underestimation could influence the main results?

Reviewer #3 (Remarks to the Author):

The authors accounted for most points raised in the previous review cycle, but one main point has not entirely covered, namely the scale dependency and the use of an equal area projection for community analyses. From authors' reply, I understand that scale-dependency is not pressing when modelling habitat suitability of individual species; I fully agree in this; the problem however is valid when the analyses are conducted at the community level, to model the co-occurrence of species; indeed, when species distributions

were aggregated to 3,000 arc seconds (~100 km) in this case the equal area projection is mandatory because species richness is strongly related to the sampling area. Thus, I rennovate my suggestion to conduct at least the community level analyses using a different projection respecting the sampling area and thus the co-occurrences of species and lineages.

Dear reviewers,

We would like to thank you for your comments in this second round of reviews and the opportunity to further revise our manuscript "Regional uniqueness of tree species composition and response to forest loss and climate change". We appreciate the time and effort you have spent assisting us with the improvement of our manuscript.

In this revision, we have addressed each of your comments. Please find below our point-by-point response. Changes are highlighted in blue in the manuscript.

Reviewer #1 (Remarks to the Author):

Overall, I think the authors have done a sufficient job with their revision. Please find below a few more minor comments.

Response: We would like to thank the reviewer for this positive feedback on our revision.

Lines 208-210: "The relative decrease in range size reflects how species ranges are restricted by historical forest cover loss and we find a median range reduction of 26.7% across all considered species"

>there are lots of things that may prevent species from occupying their fundamental niche. I don't think this can be definitively attributed to forest loss. A more cautious statement here would be appropriate. I note that they are cautious in the discussion.

Response: We agree that many factors may prevent species from occupying their fundamental niche and we do not intend to estimate the range restriction due to all of them. In the scope of the results mentioned above, we are only estimating the range restriction due to forest cover loss. We speculate that the confusion may be due to the term "realized ranges" being used in this context. Therefore, we rephrased the first sentence of this paragraph to not include this term (lines 204-206). We hope that this and the part of the discussion concerning these results will be sufficient to convey the limitations of our approach.

Lines 279: Or to historical and regional factors sensu Ricklefs.

Response: While we had already mentioned historical factors, we completed our sentence to include regional factors as well as a reference to Ricklefs (line 280).

It would be interesting to know the number of species modelled per biome.

Response: We added the number of species considered per biome in Figure 2.

Lines 306-310: "For instance, the phylogenetic similarity of India, subtropical Africa, and Australia highlight the signal of their past connection before their segregation caused by plate tectonics⁵⁷, while the similarity between temperate moist forests in North America and China is likely to indicate past connectivity of this biome followed by evolutionary conservatism⁵⁹."

>The authors are ignoring the increasing body of work showing lots of dispersal among continents since they separated. Most plant families are younger than continental splits. See classic papers by Susanne Renner and others (incl more

recent one by Gagnon et al. in *New Phytologist*). Sorry, don't have internet while writing this review to look up citations!

Response: Thank you for pointing out this oversight. We should have mentioned dispersal events. We added this in the discussion (lines 287 and 314-317).

The authors state in response doc, "Moreover, as these ordinations are in taxonomic and phylogenetic space, respectively, we cannot map the environmental vectors in the ordination space in Figure S4." This can readily be done with `ordifit` function in `vegan` package in R. This is more for their info. I am happy (enough) with the way the ordinations are now presented.

Response: We thank the reviewer for this information. After careful consideration, we decided to not add the environmental vectors to the ordination figures in the supplementary material in order to display simpler figures which will be easier to understand. However, we note the existence of this function for future work.

Lines 355-356: SE Asian forests have higher species richness than the Congo, so authors might consider rephrasing this sentence.

Response: We thank the reviewer for pointing this out. We adapted the relevant sentences in both the results and discussion sections to include South East Asian forests as well as the Amazon and the Congo (lines 219- 220 and 365).

Reviewer #2 (Remarks to the Author):

I appreciated the revision which addressed my previous comments well.

Response: We would like to thank the reviewer for the positive evaluation of our revised manuscript.

The major concern I have is about the species being able to include. As the authors said, they only kept species having more than 90 records for the further analyses. It indicated that those species are largely widely distributed. In contrast, lots of small-ranged species exist in the tropics (Gatti et al., 2022 PNAS; Guo et al., 2020 Preprint & 2022 PNAS; Liang et al., 2022 NEE), as the authors also mentioned in Line 357. The exclusion of those range-limited species in tropics could cause certain bias in the results, particular Fig.2a, which revealed that tropical moist forests hold lots of large range species. This finding is not incorrect, but could miss another peak in the small range size end. And this missing peak could be more vulnerable to future climate change and/or tree loss, that is, affecting the results of Fig. 2b & 3. This reminded me another potential drawback about the SDM compared to other methods to obtain species' distribution, even though the reviewers thought it the "main technical innovation" of the study. I admired how the authors can manage to get the SDM models done, however, given its requirement of high-quality and large-quantity of occurrences, the majority of species will be excluded, just like the case of this study.

Response: We agree that the exclusion of rare and under-sampled species is a major limitation of our approach and that it should be carefully considered when interpreting our results. Nonetheless, we were able to generate distributions of a considerable number of species at a relatively high resolution. We have added elements in the discussion to underline this limitation, how it may be addressed in future work and indicate how our results

may be biased due to the exclusion of rare and under-sampled species (lines 294-300 and 351-356).

One minor point is the validation using 23 species. From Fig. S4 and also the relevant text, it seems the ranges were not well matched. The authors wrote their estimated ranges tended to be conserved. How big this underestimation could influence the main results?

Response: We added a quantitative analysis to support this result. Additional plots in Figure S4 show the range sizes for our model results (SDM) and those of the maximum habitat suitability maps (MHS) to which they were compared, as well as the relative range sizes. These plots highlight that for all species but one, SDM ranges were smaller than MHS ranges and show the extent of this difference. Furthermore, we added a sentence in the results section indicating the average relative difference in range sizes of 36% (lines 144-145).

Reviewer #3 (Remarks to the Author):

The authors accounted for most points raised in the previous review cycle, but one main point has not entirely covered, namely the scale dependency and the use of an equal area projection for community analyses. From authors' reply, I understand that scale-dependency is not pressing when modelling habitat suitability of individual species; I fully agree in this; the problem however is valid when the analyses are conducted at the community level, to model the co-occurrence of species; indeed, when species distributions were aggregated to 3,000 arc seconds (~100 km) in this case the equal area projection is mandatory because species richness is strongly related to the sampling area. Thus, I re-iterate my suggestion to conduct at least the community level analyses using a different projection respecting the sampling area and thus the co-occurrences of species and lineages.

Response: We thank the reviewer for the positive evaluation of the previous revisions and for underlining the importance of scale dependency for our community-level analyses. We agree with the suggestion of the reviewer and, therefore, repeated the analyses of taxonomic and phylogenetic composition by sampling our modelled distributions with an equal area projection at 100 km resolution and recomputing the ordinations. We updated Figure 1 with the updated results. The results look slightly different visually, as the colors are determined by the specific values from each of the ordination axes. However, the patterns and differences between the taxonomic and phylogenetic ordinations remain similar. The relevant figures in the supplementary material were also updated (Figures S5, S6, and S7). We updated the legend of Figure 1 (lines 157-158), the relevant text in the results section (lines 167, 182, 184, 186, 188) and the methods section (lines 507-509, 518, 529-530).

REVIEWERS' COMMENTS

Reviewer #2 (Remarks to the Author):

I am very happy to see that the authors have addressed my main concerns. Thanks so much for all of your work on this.

Reviewer #3 (Remarks to the Author):

Overall, I think the authors have done a sufficient job with their revision; particularly, they sampled and modelled distributions with an equal area projection as suggested and recomputed the ordinations.

Reviewer #3 (Remarks on code availability):

The codes are reproducible and usable for the community.